# Magmas Inhibition in Prostate Cancer: A Novel Target for Treatment-Resistant Disease

**DOI:** 10.3390/cancers14112732

**Published:** 2022-05-31

**Authors:** Jianhui Yang, Bhaskar C. Das, Omar Aljitawi, Avinash Kumar, Sasmita Das, Peter Van Veldhuizen

**Affiliations:** 1Wilmot Cancer Institute, Department of Medicine, University of Rochester Medical Center, Rochester, NY 14642, USA; jianhui_yang@urmc.rochester.edu (J.Y.); omar_aljitawi@urmc.rochester.edu (O.A.); 2Arnold and Marie Schwartz College of Pharmacy and Health Sciences, Long Island University, Brooklyn, NY 11201, USA; avinash.kumar@liu.edu (A.K.); sasmitad2006@gmail.com (S.D.); 3Division of Nephrology, Department of Medicine, Icahn School of Medicine at Mount Sinai, New York, NY 10029, USA

**Keywords:** magmas, magmas inhibitor, apoptosis, stem cell, resistance, necrosis, prostate cancer

## Abstract

**Simple Summary:**

Metastatic and treatment-resistant prostate cancer remains a life-threatening disease despite recent therapeutic advances. Literature suggests that treatment resistance and prostate cancer progression is associated with prostate cancer stem cells. In this study, we evaluated the role of the mitochondria-associated granulocyte–macrophage colony-stimulating factor signaling (Magmas) protein as a molecular target and applied a novel Magmas inhibitor, BT#9, on prostate cancer cells and normal control cells. We found that Magmas was overexpressed in human prostate cancers and its expression was linked to the aggressiveness of the disease. BT#9 downregulated Magmas expression, reduced viability and induced apoptotic cell death in prostate cancer cells. The mechanism of cell death by BT#9 is mainly caspase-independent and via a Reactive Oxygen Species (ROS)-mediated pathway. This is the first study that has evaluated targeting the Magmas protein in prostate cancer and, to our knowledge, the first to elucidate the potential molecular mechanism of BT#9 activity in prostate cancer, including the mode of cell death and the critical role of ROS accumulation. Our work may provide a potential clinical application for a novel prostate cancer treatment that can overcome cancer stem cell and therapeutic resistance.

**Abstract:**

The purpose of our study was to evaluate Magmas as a potential target in prostate cancer. In addition, we evaluated our synthetic Magmas inhibitor (BT#9) effects on prostate cancer and examined the molecular mechanism of BT#9. A cell viability assay showed that treatment with BT#9 caused a significant decrease in the viability of DU145 and PC3 prostate cancer cells with little effect on the viability of WPMY-1 normal prostate cells. Western blot proved that BT#9 downregulated the Magmas protein and caspase-3 activation. Flow cytometry studies demonstrated increased apoptosis and disturbed mitochondrial membrane potential. However, the main mode of cell death was caspase-independent necrosis, which was correlated with the accumulation of mitochondrial and intra-cellular Reactive Oxygen Species (ROS). Taken together, our data suggest Magmas is a potential molecular target for the treatment of prostate cancer and that Magmas inhibition results in ROS-dependent and caspase-independent necrotic cell death.

## 1. Introduction

Prostate cancer (PC) is the most commonly occurring cancer and the second leading cause of cancer mortality in men in the United States. Approximately 248,530 men were diagnosed with PC and approximately 34,130 men died in the US in 2021 [1]. In patients with primary prostate cancer, bone metastases are by far the predominant metastatic site and the primary contributing factor to this mortality. In a study that performed autopsies in 1589 patients who died with metastatic castrate resistant prostate cancer, bone metastases were found in 90% confirming this dominant issue [2]. In metastatic prostate cancer there is a nearly 90% response to androgen deprivation therapy. Although there are currently multiple therapies targeting the androgen receptor, there is a universal disease relapse despite the encouraging and prolonged initial tumor responses [3]. Prostate cancer stem cells are believed to be integral to this ultimate disease progression. Cancer stem cells (CSCs) are a small subpopulation of cells that possess characteristics associated with normal stem cells, allowing for self-renewal and differentiation. They are integrally responsible for tumor relapses as well as the development of therapeutic resistance and may be protected by the bone marrow niche [4,5]. The bone marrow niche plays a significant role in promoting prostate cancer cell growth in the bone marrow and this relationship with the microenvironment appears to contribute to the treatment resistance of CSCs, [6]. Despite our increasing knowledge of the biology and role of CSCs and their interaction with the tumor stroma, therapeutic options which specifically target CSC are limited.

Using a novel inhibitor, we evaluated mitochondria-associated granulocyte–macrophage colony-stimulating factor signaling molecule (Magmas) as a potential therapeutic target in prostate cancer. Magmas, a highly conserved and essential protein overexpressed in aggressive prostate cancer, is an attractive novel target. Mitochondrial targets such as Magmas have been shown to have an increasing role in the development of cancer stem cells and therapeutic resistance [7,8]. Traditional chemotherapy is dependent on cell growth to be effective and the dormancy of CSCs is one of the reasons for their treatment resistance. Mitochondrial-targeted therapy, however, does not have this dependence on cell turnover.

Magmas was first identified in a study investigating granulocyte colony-stimulating factor (GM-CSF)-responsive genes. In that study, rapid induction of Magmas occurred when the human myeloid cell line PGMD1 was exposed to GM-CSF when compared to interleukin-3 [9]. Subsequent studies characterized its gene structure, confirmed its mitochondrial location, and identified its potential role in protein transport and mitochondrial biogenesis [10]. Further contributing to the knowledge and function of Magmas were experiments showing that it is overexpressed in two-thirds of pituitary adenomas. It is also highly expressed in two ACTH-secreting pituitary cell lines and loss-of-function studies demonstrated the role of Magmas in protecting these cells from apoptosis, signifying a role in cancer development [11,12]. One study examined Magmas expression by immunohistochemistry in human prostate cancer tissues obtained at the time of radical prostatectomy. Magmas was expressed in 14 out of 15 tumors, with variable expression ranging from one tumor demonstrating weak staining in 10% of tumor cells to another showing strong staining in greater than 90% of malignant cells. Normal prostate showed either no or only weak staining in a small percentage of cells whereas the premalignant lesion, prostatic intraepithelial neoplasia (PIN), had a staining pattern intermediate between the normal and malignant tissues [13]. Although this data supports the potential role of Magmas in the carcinogenic process, its role in the development of treatment resistance is still largely unknown.

As previously reported, we designed several small-molecule Magmas inhibitors (SMMI) and assayed for their effects on proliferation in yeast [14]. The most active compound, Bhaskar Technology #9(BT#9), inhibited growth at a 4 μM scale and this compound was shown by fluorometric titration to bind to Magmas with a K_d_ = 33 μM. In this this prior study, target specificity of BT#9 was established by demonstrating direct binding of the compound to Magmas [14]. Here, we evaluated Magmas expression in human prostate cancer cell lines, patient tissues, and publicly available prostate cancer patient datasets and further elucidated the anticancer effect of BT#9 on prostate cancer cells in vitro. Our specific objectives were to establish Magmas as a potential target in prostate cancer, to further delineate the mechanism of BT#9 and to investigate it as a potential target in treatment-resistant prostate cancer cells.

## 2. Materials and Methods

### 2.1. Chemicals and Reagents

BT#9 was synthesized using our previous established procedures [14]. Docetaxel, N-acetyl-l-cysteine, were purchased from Sigma (Burlington, MA, USA). Z-VAD-FMK was purchased from R&D system (Minneapolis, MN, USA). TRAIL and SMAC mimic (Nirinapanit) were purchased from PeproTech (Rocky Hill, NJ, USA) and ApexBio Tech LLC (Houston, TX, USA). MTS and PMS were purchased from Abcam (Waltham, MA, USA) and Sigma (Burlington, MA, USA). Most chemicals were dissolved in dimethyl sulfoxide (DMSO) at proper stocking solution (1000-fold of working concentration), the DMSO volume in final working concentration was equal to or below 0.1% (*V*/*V*) and had no noticeable inhibitory effects on cultured cells.

### 2.2. Cell Culture

Human prostate cancer cell lines LNCaP, PC3, DU145 and the EBV transformed prostate/stromal cell line WPMY-1 were acquired from American Type Culture Collection (ATCC, Manassas, VA, USA; catalog numbers CRL-1740, CRL-1435, HTB-81 and CRL-2854, respectively). The cell lines PC3, DU145 and WPMY-1 were maintained in Dulbecco modified Eagle medium (DMEM). RPMI-1640 medium was used for LNCaP cells. In all cases, the medium was supplemented with 10% fetal bovine serum (FBS) and 1% penicillin/streptomycin solution.

### 2.3. Patient Tissues

Human prostate tissue samples (normal adjacent and prostate tumor tissue pairs) were purchased from the Biospecimen Repository Core Facility, University of Kansas Cancer Center. The Institutional Review Board (IRB) of the Veterans Administration Medical Center, Kansas City gave approval for the use of the tissue samples purchased under the protocol named: “Effect of Magmas inhibitor in Xenograft mouse model”, ID: 00835.

### 2.4. Western Blotting

Protein was extracted from cells growing on 6-well plates with or without treatment with BT#9. Cell and tissue lysates were prepared in the lysis buffer (50 mm Tris-HCl, pH 8.0, 0.1% SDS, 150 mm NaCl, 1% Nonidet P-40, protease inhibitor mixture: 1 μg/mL aprotinin, 1 μg of leupeptin, and 1.0 mm PMSF). Equal amounts of protein were loaded to 10% SDS-PAGE and transferred to a nitrocellulose membrane by tank blotting system (Bio-Rad, Hercules, CA, USA). Membranes were incubated with anti-Magmas antibody overnight followed by HRP-conjugated secondary antibodies. The expression of the protein, β-actin, was used as a loading control. The polyclonal anti-Magmas antibody was a kind gift from Paul T. Jubinsky, M.D., Ph.D. and Mary Short, Ph.D., Albert Einstein Medical College, New York. The Caspase-3, Anti-β-actin, the Anti-rabbit IgG, HRP-linked and Anti-mouse IgG, HRP-linked antibodies were purchased from Cell Signaling Technology (Danvers, MA, USA); catalog numbers 9669, 3700, 7074 and 7076, respectively. Thermo Scientific 1-Step Ultra TMB-Blotting Solution (catalog number # 37574) was used to determine the protein band intensity.

### 2.5. MTT and MTS Cell Viability Assays

MTT assay was performed using the Vybrant^®^ MTT Cell viability Assay Kit purchased from Life Technologies (Grand Island, NY, USA, catalogue number: V13154) according to manufacturer’s protocol. Briefly, the cells were seeded in a 96-well plate at 10,000/well and underwent overnight incubation, then treated with 1–20 μM of BT9 for 24 and 48 h. MTT reagent was then added to the cells and the absorbance of the color developed was measured at 570 nm. The Statistics Calculators (www.danielsoper.com) was used to analyze the data. In addition, for direct comparison between cell viability and apoptosis rate from the same size plate, or for further validation purposes, we also ran MTS Cell Viability Assay for cell cultured on 6- or 12-well plates; the optical density at 490 nm was read by Bioteck SynergyǁMulti-Detection Microplate Reader (Winooski, VT, USA).

### 2.6. Terminal Deoxynucleotidyl Transferase (TdT) dUTP Nick-End Labeling (TUNEL) Assay

Late stage of apoptosis was quantified by TUNEL based Apo-BRDU kit (BD Biosciences, San Jose, CA, USA). Briefly, collected cells were fixed with 1% paraformaldehyde for 30 min on ice. Cells were then resuspended in ice-cold 70% ethanol and stored at −20 °C until use. Fragmented DNA was labeled by the staining buffer, including fluorescein-dUTP and TdT, for 90 min at 37 °C. After washing and centrifuge, the cell pellet was resuspended in 0.5 mL of PBS containing 5 µg/mL propidium iodide and 500 µg/mL RNase A. After 30 min of incubation in the dark at room temperature, cells were analyzed using a FACScan LSR II (BD Biosciences, San Jose, CA, USA).

### 2.7. Detection of Mitochondrial Membrane Potential (ΔΨm) by Flow Cytometry

Mitochondrial depolarization was monitored with the potentiometric dye JC-1 using the Mitoprobe assay kit (Invitrogen) according to the manufacturer’s instructions. Cells were treated with BT#9 for 4 and 24 h, then incubated in medium containing 2 µM JC-1 for 10 min. The samples were washed with PBS, resuspended in 500 μL PBS, and kept on ice until analyzed by BD LSR II cytometer. Results were further processed with Flowjo Software (Tree Star, Inc., Ashland, OR, USA). JC-1 is readily taken up by cells and healthy mitochondria, and the JC-1 probe (green fluorescence) is a monomer at low membrane potential but yields ‘Jaggregates’ (red fluorescence) at elevated membrane potential. The mitochondrial membrane potential was evaluated as the red/green fluorescence intensity ratio.

### 2.8. Immunofluorescence and Fluorescence Microscopy

Cells were grown on 4-well chamber slide (NEST Scientific, Woodbridge Township, NJ, USA) and fixed with 4% formaldehyde at room temperature for 15 min. After rinsing three times, slides were blocked by blocking buffer at room temperature for 60 min. Next, they were incubated with cleaved caspase-3 antibody (Cell Signaling Technology, Danvers, MA, USA)—conjugated Alexa Fluor 488 for 1 h at RT. Pictures were taken by Olympus (Tokyo, Japan) fluorescence microscope system, including CKX41 inverted microscope, DP74 camera and CellSense entry Software.

### 2.9. Annexin V/SYTOX Green Assay for Apoptosis and Necrosis

Apoptotic and necrosis was determined using the Dead Cell Apoptosis Kit with Annexin V- APC and SYTOX^®^ Green for flow cytometry (Life Technologies, Grand Island, NY, USA; catalogue number: V35113) according to manufacturer’s protocol. Briefly, apoptotic cells were detected by the binding of annexin V to externalized phosphatidylserine. Annexin V+/SYTOX Green- were considered early apoptotic death. The late apoptotic and necrotic cells have non-intact membranes that permit SYTOX^®^ Green stain entrance to cellular nucleic acids. These populations of cells were detected using the Flow Cytometry Core Laboratory at The University of Kansas Medical Center.

### 2.10. Detection of Mitochondria and Intracellular ROS Production

MitoSOX red mitochondrial superoxide indicator (Invitrogen, Waltham, MA, USA) was used to measure the presence of mitochondria superoxide. The procedure was conducted according to the manual with slight modification. Before use, MitoSOX Red reagent was dissolved in DMSO 13 µL to yield a 5 mM stock solution. After BT#9 treatment at desired time, 1 μL of stocking solution was added to medium of 100 µL in an Eppendorf tube and vortexed, then added directly to 1 mL medium in a 12-well plate with a final working concentration of 2.5 μM. After incubation at 37 °C for 20 min, analysis was conducted by both fluorescence microscopy and BD LSR II cytometer (green laser with emission wavelength at 575 nm, PE channel). In addition, analysis of intracellular ROS generation was evaluated by adding a fluorescent CellROX Green Reagent (Invitrogen, Waltham, Massachusetts), followed by flow fluorescence microcopy and cytometry. Briefly, cells were plated in 12-well plates at a final density of 2 × 10^5^ cells/well and incubated overnight. Cells were then treated with the BT#9 at concentration of 10 µM or 20 µM for 24 h. Cells were stained with 2.5 μM CellROX Green Reagent for 20 min at cell culture incubator. Finally, analysis was conducted by both fluorescence microscopy and BD LSR II cytometer (blue laser with emission wavelength at 515 nm, FITC channel).

### 2.11. Dataset Analysis

Prostate cancer patient datasets available in the Oncomine and cBioPortal databases were queried for Magmas (TIMM16, PAM16) mRNA overexpression and copy number alterations, respectively.

### 2.12. Drug Interaction Relationships Analysis

Drug combination effects were determined using Compusyn analytical (version 1.0) software (T. C. Chou and N. Martin, Memorial Sloan-Kettering Cancer Center, New York and were expressed as a combination index (CI). CI is a quantitative measure of the degree of drug interaction. CI < 0.9, CI = 0.9–1.1, or CI > 1.1 represents synergistic, additive, and antagonistic relationships, respectively.

### 2.13. Statistical Analysis

Data are expressed as the mean ± standard deviation. All statistical analyses were performed using GraphPad Prism software (version 7.0, La Jolla, CA, USA). One-way analysis of variance and Tukey’s tests were performed for multiple group comparisons. * *p* < 0.05 was considered statistically significant.

## 3. Results

### 3.1. Magmas Is Overexpressed in Human Prostate Cancer Cell Lines and Prostate Tissues

To determine the Magmas expression in human prostate cancer, we examined the Magmas protein levels in a panel of prostate cancer cell lines in comparison to normal prostate cells. We found that Magmas protein is expressed at significantly elevated levels in the prostate cancer cell lines LNCaP, PC3 and DU145 as compared to WPMY-1, the EBV-transformed normal prostate/stromal cell line (Figure 1A,B). Along the same lines, we determined Magmas expression in human prostate tumor tissue biopsies in comparison to normal prostate tissues adjacent to the known tumor and found that prostate tumors exhibited significantly elevated levels of Magmas protein in comparison to adjacent normal prostate tissue in four of five cases (Figure 1C,D).

### 3.2. Bioinformatics Analysis Based on Multiple Tissue Database Further Support Magmas Is Overexpressed in Human Prostate Cancer

To strengthen our findings from smaller patient sample numbers, we interrogated the Oncomine database to determine the Magmas mRNA expression in prostate cancer. Out of the eleven, eight prostate cancer expression profile datasets [15,16,17,18,19,20,21,22] clearly showed significant overexpression of Magmas in prostate tumors relative to normal samples. Two datasets with large sample numbers [16,20] showed significant overexpression of Magmas in human prostate cancer (Figure 2A,B, Appendix A). The Taylor et al. dataset [20] also revealed that Magmas overexpression was strongly increased upon tumor progression to a higher Gleason score (Gleason score > 7) (Figure 2C, Appendix A), indicating that Magmas overexpression is linked to aggressiveness of prostate cancer. In addition, analysis of copy number alterations (CNA) data of the prostate cancer patient samples available from cBioPortal showed amplification of Magmas (Figure 2D, Appendix A), and a strong positive correlation between Magmas copy number and mRNA expression (Figure 2E, Appendix A) in five datasets [23,24,25,26,27]. Taken together, our data provide substantial evidence for involvement of Magmas in prostate cancer pathophysiology and its usefulness as a novel pharmacological target in prostate cancer.

### 3.3. BT#9 Downregulated Magma’s Expression, Reduces Viability in Multiple Prostate Cancer Cells but Spare Normal Prostate Cancer Cells

In order to observe the on-target effect, we ran a Western blot to measure the expression of Magmas protein after BT#9 treatment at the concentration of 1 µM and 10 µM. Figure 3A shows the chemical structure of BT#9. As Figure 3B and Appendix A shows, 10 µM BT9 significantly downregulated Magmas protein of androgen independent DU145 cells, but 1 µM did not change it significantly. In addition, MTT showed reduced cell viability of DU-145 after treatment BT#9(10 or 20 µM) for 24 and 48 h compared with the DMSO vehicle control (Figure 3C), but 1 µM did not reduce significantly, which was consistent with the no protein downregulation by 1 µM BT#9 as the Western Blot result shown. Similar effects on cell viability were seen in both androgen independent cell line PC3 (Figure 3D) and the androgen dependent cell line LNCaP (Figure 3E). Interestingly, BT9 did not affect the cell viability of WPMY-1 normal prostate cells relative to the DMSO vehicle control either at 24 or 48 h (Figure 3F).

### 3.4. BT#9 Disrupts Mitochondria and Induces Apoptosis, but the Main Mode of Cell Death Was Caspase-Independent Necrosis

Cell viability assays quantify the amount of cell death but do not delineate the mode of cell death after BT#9 treatment. Most published research about the role of Magmas in cell death is about apoptotic suppression [11,12]. The pan-caspase inhibitor Z-VAD-FMK inhibits caspases by irreversibly binding to their catalytic site and is a key compound for studies on apoptosis. Interestingly, Z-VAD-FMK (20 μM), failed to protect BT#9 (10 μM or 20 μM)-induced cell deaths (*p* > 0.05) (Figure 4A and Appendix A). However, Z-VAD-FMK almost completely suppressed cell death of PC-3 cells when treated by the combination of 100 ng/mL TRAIL and 1 μΜ SMAC mimic(T&S), which are known apoptosis-inducing reagents and served as positive controls. Therefore, we hypothesized that BT#9-induced cell death might be involved in apoptosis but mainly in the caspase-independent pathway.

Magmas has an essential role in the mitochondrial protein import motor [28]; we presumed that BT#9 might directly or indirectly disturb mitochondrial membrane potential (MMP, ΔψM) after binding and degrading the Magmas protein. Flow cytometry analysis in combination with a fluorescent probe such as JC-1 is widely used to monitor mitochondrial MMP, which is related to cell death, the main intrinsic pathway of apoptosis. We found a significantly decreased red/green fluorescence ratio 24 h after BT#9 treatment at both 10 and 20 µM in DU-145 and PC-3 cells (*p* > 0.05), and the earliest detected change was only 4 h after 20 μM BT#9 treatment (Figure 4B and Appendix A), indicating mitochondrial depolarization might be an early event.

As mentioned earlier, caspases are crucial mediators of apoptosis; compared with most other caspases, caspase-3 is a frequently activated executive caspase and easily detected by Western blotting because of its abundant expression. Although we did not capture significant cleaved caspase-3 after treatment with 10 µM BT#9 in 24 h or earlier time, the Western blot detected a slightly decreased level of total caspase-3 (Figure 4C and Appendix A), which was an indirect sign of caspase-3 activation. In addition, this caspse-3 activation was also confirmed by immunofluorescence microscopy (Figure 4D and Appendix A). However, compared with T&S, only limited cleaved caspase-3 and cleaved PARP were detected after treatment with BT#9 in PC-3 cells.

The TUNEL assay was designed to detect apoptotic cells that undergo DNA degradation during the late stages of apoptosis. In order to get a direct and perhaps more accurate comparison of the cytotoxicity and apoptosis assay, 2 × 10^5^/well cells were seeded in a duplicated 6-well plate and treated by 10 μM BT#9 or T&S for 48 h. The apoptosis rates of PC-3 cells treated by 10 µM BT#9 were about 11.8%, much lower than cytotoxicity, about 46.7% (Figure 4E and Appendix A), indicating another mode of cell death might be predominant. Necrosis can be detected by measuring the permeability of the plasma membrane to a normally impermeable fluorescent dye, such as the propidium iodide (PI). We ran PI staining followed by fluorescence microscopy and found dramatic cell death when treated by BT#9 at a concentration of 10 or 20 μΜ (Figure 4F and Appendix A), which is proportional to the significantly decreased cell viability treated by BT#9. The Annex V assay revealed more necrosis than apoptosis after BT#9 treatment (Figure 4G and Appendix A), which further supported our hypothesis that a non-caspase-dependent necrosis, instead of caspase-dependent apoptosis, is the main mode of cell death.

### 3.5. ROS Accumulation Is a Critical Mechanism of BT#9-Induced Cell Death

Why targeting Magmas by BT#9 induced significant necrosis instead of apoptosis is unclear; even the function of Magmas in cell death is not fully elucidated. Besides the intra-mitochondria location of Magmas and its role in mitochondrial protein transport, it was also reported that Magmas functions as a negative ROS regulator [29]. Taken together, it is reasonable to hypothesize that excessive reactive oxygen species (ROS) is possibly induced and involved in necrosis after BT#9 treatment. Indeed, under oxidative stress, excessive ROS induce cellular damage and cell death.

Mitochondria is the main source to generate ROS, mainly superoxide and hydrogen peroxide. We first detected mitochondrial ROS by MitoSOX Red superoxide indicator, a mitochondrial-localizing dye. After 8 h of treatment, 10 μM and 20 μM BT#9 induced significant ROS accumulation as reflected by increased fluorescence intensity under a microscope (Figure 5A and Appendix A), indicating markedly increased superoxide, a major species of ROS. The increased ROS is visible but to a lesser extent after BT#9 treatment at the concentration below 10 μΜ, and this low-level intra-mitochondria ROS accumulation is proportional to limited cytotoxicity measured by a cell viability assay. Therefore, BT#9-induced intra-mitochondria superoxide accumulation is dose-dependent. Meanwhile, samples were harvested immediately after the observation under fluorescence microscope for FCM analysis and further quantification purpose. Although the MitoSOX Red superoxide indicator kit is mainly designed for fluorescence microscope, published literature explored its application in flow cytometry [30,31] in which the percentages of positive cells were quantified by proper gating. As Figure 5B and Appendix A shown, the histogram of flow cytometry showed about 11% or 30% of cells were positive about 8 h after 10 μM or 20 μM BT#9 treatment in PC-3 cells, respectively.

We next set out to assess the production of intracellular total ROS by CellROX Green Reagent. The generation of total ROS can bind to DNA upon oxidation and turn the nuclei DNA green. Both fluorescence microscopy (Figure 5C and Appendix A) and FCM (Figure 5D and Appendix A) showed dramatic accumulations of ROS 24 h after BT#9 treatment. Meanwhile, fluorescence microscopy proved that antioxidant reagent, 100 µM N-acetyl L-cysteine (NAC), markedly inhibited the ROS accumulation after 10 or 20 μM BT#9 treatment in PC-3 cells. Then, MTS cell viability assay shown that 100 or 1000 µM NAC significantly protected against 10 or 20 μM BT#9 induced cell death in a dose-dependent manner (*p* < 0.05) (Figure 5E, Appendix A-2 and Appendix A). In summary, induced ROS is proposed as a critical mechanism of BT#9 induced cell death.

### 3.6. Combination Effect of BT#9 and TRAIL and Docetaxel

In view of the direct effect of BT#9 on MMP, we hypothesized that BT#9 might also act as a potent or mild “apoptosis regulator” other than a mild “apoptosis trigger”. Next, we investigated the combination of lower concentration (sub-effective) of BT#9 and TRAIL on a TRAIL resistance PC-3 cells. As MTS cell viability assays shown (Figure 6A, Appendix A and Appendix A), there is a statistical difference between TRAIL (10 ng/mL or 100 ng/mL) and the combination of TRAIL (10 ng/mL or 100 ng/mL) and 2 μΜ BT#9 (*p* < 0.05). The drug combination index (CI) between TRAIL (10 ng/mL or 100 ng/mL) and BT#9 (1, 2 and 5 μM) is 0.22–0.72 (Figure 6B and Appendix A), which indicates the existence of synergism. Docetaxel is a current first-line chemotherapeutic drug for hormone refractory prostate cancer, and inducing apoptosis was regarded an important mechanism to exert its cancer-cell-killing effect. When concurrently treating the PC-3 cells with BT#9 (1,2 and 5 μM) and docetaxel (0.1 μM or 1 μM), there was no statistically difference between docetaxel (0.1 μM or 1 μM) and combined treatment (*p* > 0.05) (Figure 6C, Appendix A-4 and Appendix A), with an additive effect revealed by combination effect analysis (CI = 0.86–1.06, Figure 6D and Appendix A). In addition, we found that the expression of Magmas of PC-3 cells and DU-145 cells was upregulated by docetaxel treatment as Figure 6E shown. Theoretically, BT#9 might become more efficacious when Magmas upregulated. Therefore, whether sequential treatment, docetaxel followed by BT#9, has synergistic effect also needs examination. In our study, the PC-3 cells were pre-treated with docetaxel (0.1 and 1 μM) for 48 h, cell culture media was replaced with new media containing BT#9 (1, 2, or 5 μΜ). The results show that the CI for docetaxel (0.1 and 1 μM) and BT#9 (1 or 2 μΜ) was still around 1 (0.95–1.15), but docetaxel (0.1 and 1 μM) and BT#9 (5 μΜ) was 0.39–0.45 (Figure 6F and Appendix A).

## 4. Discussion

Mitochondria play a critical role in cancer bioenergetics and are considered potential targets for cancer treatment [32]. Current mitochondria-targeting anticancer drugs still have limited efficacy, so it is imperative to develop novel and potent mitochondria-targeting agents. Based on the reported observations that Magmas may be a viable cancer therapeutic target, we developed the first reported Magmas inhibitor, BT#9 [14], which is now being validated by other investigators in other malignances [33,34], but our report is the first exploring the effect of BT#9 on prostate cancers. MTT assays found that BT#9 reduced significant cell viability in multiple prostate cancer cell lines including PC3, DU145 and LNCaP in a dose-dependent and time-dependent manner, but interestingly, BT#9 spared normal prostate cancers WPMY-1. The underlying mechanism might be contributed to lower expression of Magmas in normal WPMY-1 cells. In terms of the specific binding of BT#9, it was reported that BT#9 can bind Magmas protein as fluorometric titration and molecular modeling has shown [14]. Our protein expression analysis by Western blot further confirmed that the Magmas protein was decreased dramatically by BT#9 at the concentration of 10 µM but not 1 µM after 24 h. The insufficient downregulation of Magmas protein by 1 µM BT#9 is consistent with insignificantly reduced cell viability at 24 h.

Mitochondria is not only indispensable for energy production of cancer cells, but is also the critical regulator of the mitochondrial pathway of apoptosis. In view of the intra-mitochondria location of Magmas and its role of apoptosis inhibition as mentioned before, we initially expected that targeting mitochondria protein Magmas can induce predominant apoptotic cell death via mitochondria apoptosis, which is the tentative main mode of cell death of prostate cancer cells after treatment with BT#9 at the concentration of 10 or 20 µM. Unexpectedly, when applied pan-caspase inhibitor Z-VAD-FMK, the 10 or 20 µM BT#9 treated cell death was not protected significantly. Therefore, we presume that activation of caspase or apoptosis induction might not be a critical event for cell death induced by BT#9.

It is widely accepted that under the cell death scenario, that the loss of MMP is a hallmark of intrinsic apoptosis and an upstream event of caspase activation in the intrinsic pathway. JC-1 is considered the gold standard for the evaluation of the loss of MMP. We ran JC-1 fluorescence assay and detected decreased red/green fluorescence ratio as earlier as 4 h after BT#9 (20 μM) treatment, which is a sign of decreased MMP and an early event of apoptosis activation. In addition, the loss of MMP is presumed to be accompanied with the opening of the mitochondrial permeability transition pore (MPTP), allowing small molecules (such as cytochrome c) to be released into the cytosol where caspases will be activated. Caspase-3 is the main executive caspase in apoptosis. In our study, both Western blot and immunofluorescence microscopy demonstrated limited activation of caspase-3, which explained why the caspase inhibitor failed to protect BT#9 induced cell death. In addition, Z-VAD-FMK only inhibits multiple caspases activation, but it may not inhibit caspase-independent apoptosis. The concept of caspase independent apoptosis is still under debate. In spite of this uncertainty, we needed to identify and quantify the apoptosis rate by the TUNEL based apoptosis assay, which exclusively detects the characteristic event in later stages of apoptosis.

Necrosis has been classically defined as an unprogrammed form of cell death that occurs in response to overwhelming chemical or physical insult. Fluorescence microscopy after PI staining proved dramatic cell death 48 h after BT#9 treatment (Figure 4F), and Anexin V assay (Figure 4G) further proved the cell death was a mode of more necrosis less apoptosis. To sum up, BT#9 caused significant cytotoxicity, might devastate cancer cells and mainly caused necrosis, while apoptosis was only activated to a low level due to an unknown suppressed mechanism.

As mentioned earlier, Srivastava et al. [29] have shown that Magmas may function as a regulator of ROS and protects cells against oxidative stress-mediated damages. Magmas protein promotes cellular tolerance toward oxidative stress by enhancing antioxidant enzyme activity, thus preventing the induction of apoptosis. Investigators in their study showing siRNA-mediated knockdown of Magmas resulted in a more than 1.5-fold increase in ROS levels, while overexpression of Magmas rescues the viability of these cancer cells from oxidative stress. Therefore, we intended to observe and measure the mitochondrial and intercellular ROS after BT#9 treatment. In this study, we found generation of ROS after BT#9 treatment detected by two ROS detection assays which compensate and validate each other. The first one detects mitochondria ROS but is limited to superoxide only, the second one detects intracellular total ROS but cannot distinguish an intra-mitochondria or extra-mitochondria source. When results are combined, we conclude that BT#9 definitely induced intracellular ROS accumulation, with a possibility of BT#9 targeting mitochondria and inducing intra-mitochondria ROS initially. Because in our dynamic observation, we found the intra-mitochondrial fluorescence signal reached the strongest signal around 8 h as we selected this peak time for data record, while the strongest signal intracellularly signal was around 24 h. Under the same experimental condition, NAC in combination not only suppresses ROS accumulation remarkably but also significantly protected BT#9 induced cell death, indicating the pivotal role of ROS in BT#9 induced cell death. The dramatic abrogation of cell death in prostate cancer cells treated by BT#9 suggests that ROS is a possible mediator of cell death. We also found that the intracellular ROS induced by BT#9 at the concentration lower than 10 μΜ is limited, which is proportional to weak cytotoxicity measured by either MTT or MTS. This provides another support for the notion that the ROS-dependent pathway is a critical mechanism of the anti-cancer effect of BT#9.

Numerous reports showed cancer cells exhibit a higher ROS level than normal cells for metabolic purposes, potential anticancer therapy can kill cancer cells by pushing cancer cells beyond a breaking point where the intracellular ROS level can damage cells. However, the death mode, conduct pathway and detailed mechanism of ROS-induced cell death is still not fully understood. What we know is that under different scenarios, ROS may induce apoptosis, necrosis, autophagy or mixed modes of cell death, depending on the cell death trigger, cell type and many other conditions. In this study, ROS not only induced cell apoptosis, but more importantly necrosis. As some research has shown [35], higher level of ROS may paradoxically inhibit the function of caspase and apoptosis. This finding may partly explain why apoptosis was not predominant in BT#9 induced cell death. Indeed, the mode of cell death still needs further investigation.

As mentioned earlier, Magmas protected ACTH-secreting pituitary adenoma from apoptotic stimuli such as staurosporine [11], indicating its negatively regulating role in response to pro-apoptotic stimuli. As a Magmas inhibitor, we also hypothesized that BT#9 might act as “apoptosis regulator” rather than “apoptosis trigger”. So far, there is no report about the combination of BT#9 and any anticancer treatments; whether synergism exist or not is unknown. TRAIL selectively induces apoptotic cell death in a variety of cancer cells and is regarded as a therapeutic apoptosis inducing treatment in some clinic trials, but TRAIL resistance is prevalent in many solid cancers including prostate cancer. Therefore, we firstly tested the combination of TRIAL and sub-effective concentration of BT#9. Results of calculated CI value (0.22–0.72) proved that a slight synergistic effect existed, while positive control SMAC mimic has a much stronger synergistic effect on TRAIL. In clinic, docetaxel retains a key role in the treatment for metastatic hormone-refractory prostate cancer (HRPC), however, almost all patients develop drug resistance after treatment. We examined the combination effect between docetaxel and BT#9 but a CI of around 1 proved the combination is just additive. Interestingly, we did not prove a synergistic effect between docetaxel and SMAC mimic either, possibly because SMAC mimic is a potent apoptosis regulator but docetaxel-induced cell death might not be apoptosis-predominant at the tested doses, which may also explain why BT#9 synergized with TRAIL but not docetaxel in concurrent treatment. As mentioned earlier, the Magmas protein was originally identified as a protein involved in GM-CSF signaling [9], we found that Magmas was also induced by anticancer treatment and other cell stress conditions. Western blot found Magmas protein in PC-3 cells was significantly upregulated by treatment with 1 μΜ docetaxel at 24 h and 48 h. It is too early to ascertain whether this inducible expression of Magmas is involved in the mechanism of drug resistance, but we did evaluate the sequential treatment of docetaxel and BT#9. We did not find a synergistic effect between docetaxel (0.1 or 1 μM) and BT#9 (1 or 2 μM), which is similar to the finding with concurrent treatment. Interestingly, there is slight synergistic effect when BT#9 is dosed at 5 μM as Figure 6E shows. The molecular mechanism for this synergism needs further investigation.

Figure 7 summarizes the potential molecular mechanism of prostate cancer cell death induced by BT#9. In brief, BT#9 has an on-target effect on the function and the expression level of Magmas protein, followed by mitochondria disturbance. Both effects further cause ROS accumulation and then activate caspase-independent necrosis, which is the predominant mode of cell death after BT#9 treatment. In addition, mitochondrial disturbance causes intrinsic apoptosis but not dramatically possibly because caspase was suppressed by excess ROS paradoxically. In addition, ROS accumulation and mitochondrial disturbance may affect each other. Further studies are being planned to understand BT#9 mechanisms of prostate cancer cell death. Regardless, BT#9 is a potent Magmas inhibitor because it caused significant cell death at the concentration of 10 µM-20 µM in vitro. The real potential clinical application is to target treatment-resistant prostate cancer cells, including prostate cancer stem cells (PCSC). Mitochondria are the most prominent source of intracellular reactive oxygen species (ROS) and low levels of ROS have been implicated in cancer cell stemness. Mitochondrial-targeted redox-active agents [32] may therefore provide a novel strategy to selectively target prostate cancer stem cells, with BT#9 being a potential candidate.

Two other published studies have evaluated BT#9 in other malignancies, one being ovarian carcinoma. Investigators found that BT#9, reduced the viability of an ovarian carcinoma carboplatin-resistant cell line (OV9) significantly more than the parental OV90 cell line, suggesting a potential role for Magmas inhibition in overcoming chemotherapeutic resistance [33]. Their data have also shown that ovarian cancer cells after treatment with chemotherapy are enriched in a population of cells having CSC-like characteristics. Magmas overexpression has also been reported in human gliobastoma (GBM) resection samples and in tumors derived from the syngenetic subcutaneous injection of the GL261 murine glioblastoma cell line [34]. In these experiments, BT#9 decreased proliferation in human GBM cell lines (D-54 MG, U-251 MG), murine embryonal stem cell lines (1123 Mes, 83 Mes) and a human glioblastoma stem cell line (HuPuP01). These published studies further support that Magmas inhibition is a viable treatment strategy, targeting both chemotherapeutic resistant and cancer-like stem cells.

Our experiments were designed to determine whether Magmas inhibition has a potential role in the treatment of prostate cancer. We initially verified Magmas protein expression, which was present in both the WPMY-1 (the EBV transformed prostate/stromal cell line) used as control and the prostate cancer cell lines. The expression, however, was less in the EBV transformed prostate/stromal cell line. We then analyzed the Magmas protein expression in fresh adjacent normal prostate tissue and prostate tumor samples. Although the differences in expression were variable, in four of five samples, the prostate cancer adjacent normal tissues showed decreased expression of Magmas protein relative to the prostate cancer. With this method of tissue collection, prostate tissues neighboring the area of the tumor frequently contain premalignant changes, which may increase Magmas expression thereby impacting the magnitude of difference. Future human studies are needed to differentiate the change in Magmas expression across the spectrum of a normal prostate gland to glands affected purely by benign prostate hyperplasia, premalignant change, and metastatic disease, and further determine how expression changes with the development of treatment resistance.

## 5. Conclusions

Collectively, Magmas is a novel therapeutic target in the treatment of prostate cancer and BT#9 reduced protein expression of Magmas and decreased the cell viability of prostate cancer cells. Our findings suggest that the main mechanism of BT#9-mediated prostate cancer cell death is increased ROS and non-caspase-dependent necrosis, while mitochondria disturbance and apoptosis might be involved to a lesser extent. In addition, the combination of BT#9 and TRAIL has a synergistic effect. Future studies will focus on continued delineation of the mechanism of BT#9-induced necrosis. In addition, this mechanism of action of BT#9 needs to be confirmed in vivo. Finally, we will also examine the effect of BT#9 on treatment-resistant prostate cancer and on prostate cancer stem cells.

## Figures and Tables

**Figure 1 cancers-14-02732-f001:**
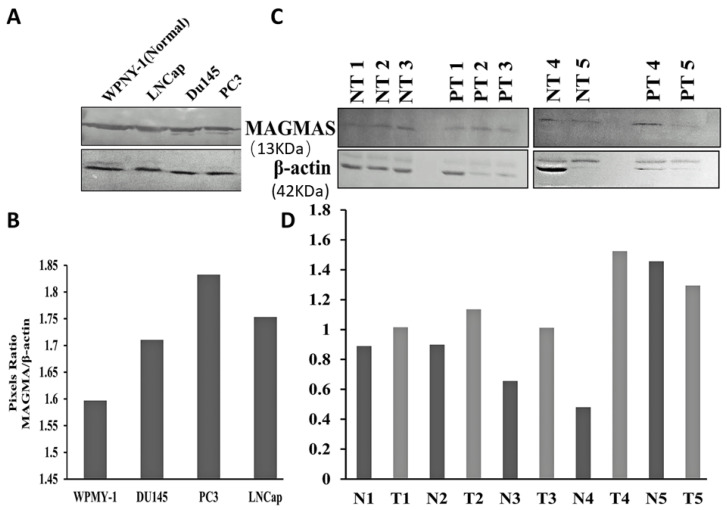
Magmas is overexpressed in prostate cancer cell lines and prostate tumor tissues. (**A**) Representative immunoblot showing Magmas protein expression in WPMY-1 (the EBV-transformed prostate/stromal cell line) and LNCaP, DU145 and PC3 prostate cancer cell lines. β-actin was used as loading control. (**B**) The intensity ratio of Magmas protein/β-actin for WPMY-1, LNCaP, DU145 and PC3 cell lines. (**C**) Representative immunoblot showing Magmas protein expression in five normal prostate cancer adjacent tissues (NT1-NT5) and prostate tumor tissues (PT1-PT5). β-actin was used as loading control. (**D**) The intensity ratio of Magmas protein/β-actin) for the five normal prostate cancer adjacent tissues (NT1-NT5) and the five prostate tumor tissues biopsies (PT1-PT5).

**Figure 2 cancers-14-02732-f002:**
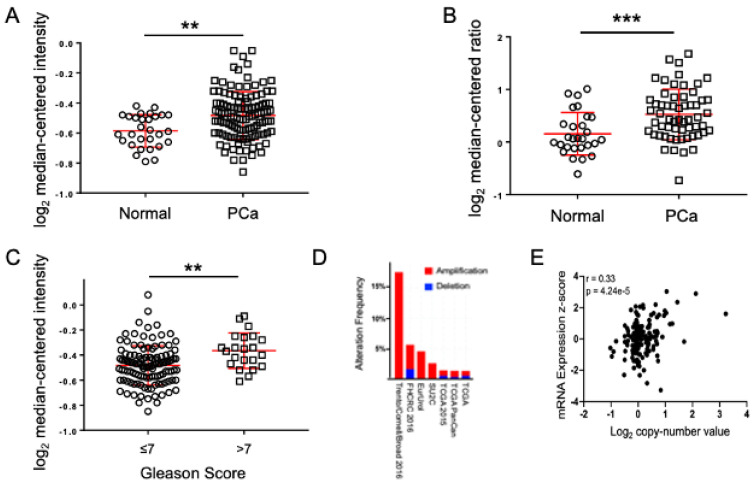
Magmas is overexpressed in human prostate cancer. (**A**,**B**) Meta-analysis showing Magmas mRNA overexpression in prostate tumor samples compared to normal tissues from expression arrays [16,20] using the Oncomine database. ** *p* < 0.001, *** *p* < 0.0001 (Unpaired two-tailed *t* test). (**C**) Meta-analysis showing increase in Magmas mRNA expression with increase in Gleason score from the expression array [16] using the Oncomine database. ** *p* < 0.001 (Unpaired two-tailed *t* test). (**D**) Copy number analysis (CNA) for Magmas in publicly available prostate cancer patient datasets [23,24,25,26,27] using cBioPortal for Cancer Genomics. (**E**) Correlation analysis for Magmas copy number and mRNA expression in publicly available prostate cancer patient datasets [23,24,25,26,27] using cBioPortal for Cancer Genomics.

**Figure 3 cancers-14-02732-f003:**
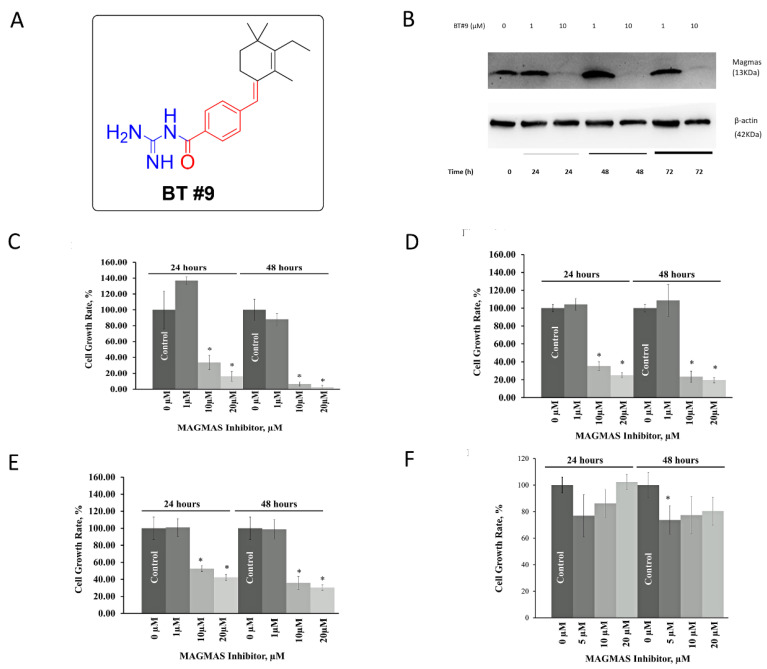
The effect of BT#9 on the protein expression of Magmas and the cell viability of prostate cancer cells and normal prostate cells. (**A**) Chemical Structure of Magmas Inhibitor BT#9. (**B**) Magmas protein in DU-145 cells was significantly downregulated by BT#9 (10 µM) at 24,48 and 72 h. (**C**) MTT assay for the cell viability analysis of DU145 cells treated with various concentrations of BT9 (0, 1, 10 and 20 µM) for 24 and 48 h. (**D**) MTT assay for the cell viability analysis of PC3 cells treated with various concentrations of BT9 (0, 1, 10 and 20 µM) for 24 and 48 h. (**E**) MTT assay for the cell viability analysis of LNCaP cells treated with various concentrations of BT9 (0, 1, 10 and 20 µM) for 24 and 48. (**F**) MTT assay for the cell viability analysis of WPMY-1 cells treated with various concentrations of BT9 (0, 5, 10 and 20 µM) for 24 and 48 h. Each experiment was repeated three times. * *p* < 0.05 (Unpaired two-tailed *t* test).

**Figure 4 cancers-14-02732-f004:**
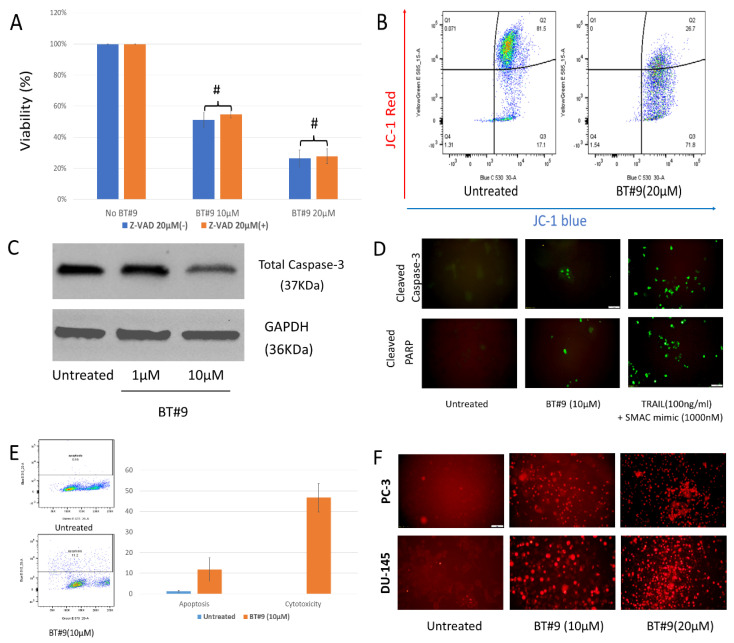
The mode of cell death treated by BT#9. (**A**) MTS viability assay showed that general caspase inhibitor (Z-VAD) did not protect BT#9-induced cell death in PC-3 cells (#: *p* > 0.05). (**B**) Dot plot of fluorescence shift in PC-3 cells after being treated with 20 μM BT#9 for 4 h. The ratios of red/green (Q2/Q3) were decreased from 81.5/17.3 (untreated) to 26.7/71.8 (treated), indicating the loss of MMP. (**C**) PC-3 cells were treated by 1 µM or 10 µM BT#9 for 24 h; Western blot showed decreased total caspase-3, indicating the activation of caspase-3. (**D**) Fluorescence microscopy also confirmed that smaller percentage of cleaved caspase-3 and cleaved PARP in PC-3 cells treated by 10 μM BT#9 when compared with positive control. (**E**) TUNEL based Apo-BrdU apoptosis rate and MTS cell viability assay treated by BT#9. (**F**) PI staining by fluorescence microscopy; dramatic cell death treated by BT#9 at the concentration of 10 or 20 μΜ were detected. (**G**) Annexin V/SYTOX Green Assay for DU-145 cells treated by BT#9(10 μM). Each experiment was repeated three times. # *p* > 0.05 (Unpaired two-tailed *t* test).

**Figure 5 cancers-14-02732-f005:**
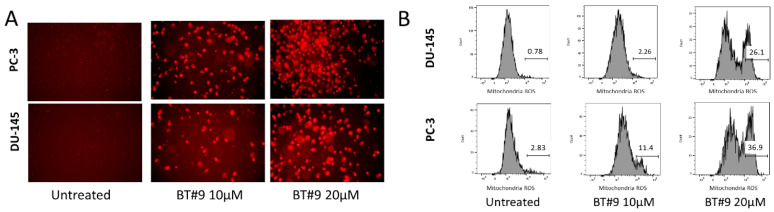
The accumulation of ROS in Prostate cancer cells after BT#9 treatment. (**A**) Intramitochondria ROS of PC-3 and DU-145 cells detected by MitoSOX Red superoxide indicator after 10 µM or 20 µM BT#9 treatment. (**B**) Representative flow cytometry histograms of MitoSOX Red fluorescence (530 nm lasor, PE channel) in PC-3 and DU-145 cells treated by 10 μM and 20 μMBT#9. Percentage of MitoSox Red positive cells was analyzed and quantified by FlowJo Software. (**C**) PC-3 and DU-145 cells were treated by BT#9 for 24 h, stained by 5 µM CellROX Green Reagent, and Intracellular ROS was detected by fluorescence microcopy. (**D**) FCM measured the intracellular ROS level in both PC-3 and DU-145 cells, showing significant ROS accululation after BT#9 treatment (20 µM). (**E**) NAC protected BT#9 induced cell death in a dose dependent way. Each experiment was repeated three times. * *p* < 0.05 (Unpaired two-tailed *t* test).

**Figure 6 cancers-14-02732-f006:**
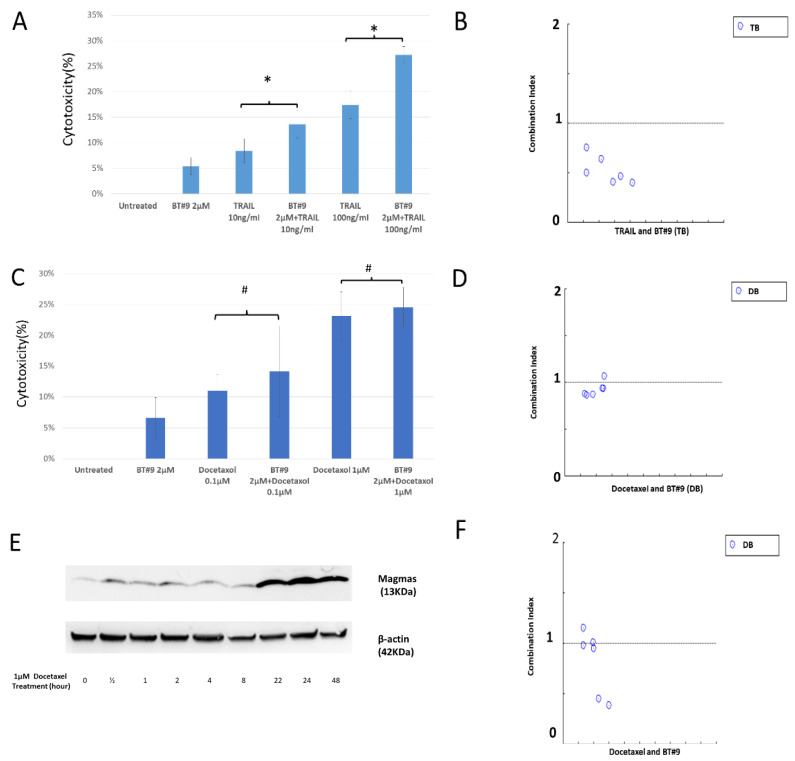
The combination of BT#9 and TRAIL or Docetaxel. (**A**,**B**) The representative result of the combination of BT#9 (2 μΜ) and TRAIL (10 ng/mL or 100 ng/mL). The whole CI of BT#9 (1, 2 or 5 μM) and TRAIL (10 or 100 ng/mL) is between 0.22 and 0.72. (**C**,**D**) The representative result of the combination between BT#9 (2 μΜ) and Docetaxel (0.1 μM or 1 μM) The whole CI of the concurrent combination of BT#9 (1, 2 or 5 μM) and Docetaxel (0.1 or 1 μM) is between 0.86 and 1.06. (**E**) The Magmas expression of PC-3 cells was upregulated around 24 h or later after 1 μM Docetaxel treatment. (**F**) The whole CI of the sequential combination of BT#9 (1, 2 or 5 μM) and Docetaxel (0.1 or 1 μM) is between 0.39 and 1.15. Among all 6 dots, upper 4 dots (CI = 0.95–1.15) represent CI for Docetaxel (0.1 or 1 μM) and BT#9 (1 or 2 Μμ), lower 2 dots (CI = 0.39–0.45) represent CI for Docetaxel (0.1 or 1 μM) and BT#9(5 μM). Each experiment was repeated three times. # *p* > 0.05, * *p* < 0.05 (Unpaired two-tailed *t* test).

**Figure 7 cancers-14-02732-f007:**
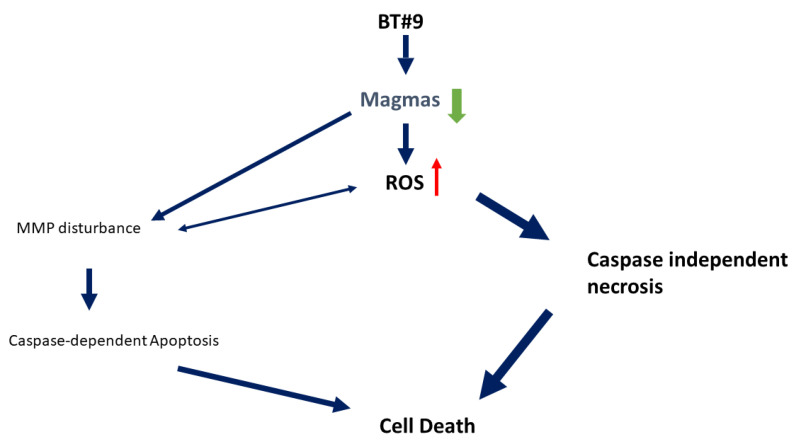
Schematic mechanism of BT#9 in prostate cancer cells. BT#9 downregulates Magma’s protein and further activated mitochondria and intracellular ROS. ROS disturbs mitochondria membrane potential and induces apoptosis, but the main cell death of BT#9 induced cell death is caspase-independent necrosis.

## Data Availability

Data is contained within the article or Appendix A.

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
