# Peer review of "Magmas Inhibition in Prostate Cancer: A Novel Target for Treatment-Resistant Disease"

_cancers, 2022, doi:10.3390/cancers14112732_

Round 1

Reviewer 1 Report

The way how the text is distributed between “Simple summary” and “Abstract” is redundant and confusing.

Where is written “Bone metastases are by far the predominant…”, please clarify if the primary site of cancer is bone. In this case, how does it enter in the presented statistics of prostate cancer? As presented in the paper? Again, the sentence  “maintaining tumor-initiating prostate cancer in the bone marrow” is confusing.

Clarify if androgen receptor is the membrane receptor in cells.

CSCs, dormancy, therapeutic resistance, and its relations with Magmas should be better explained. How does these information’s lead the authors to choose Magmas as a target?

The “role of Magmas in protecting these cells from apoptosis” should be clarified, when considering that apoptosis is a protective process that can intercept cancer in vivo;  apoptosis rate was determined in vitro. Please extrapolate the interpretation of these apoptotic events  from in vitro to in vivo.

All the text “ Therefore, we designed several small molecule Magmas…” until the end of introduction, includes methods and results. It must be elsewhere. Objectives should be stated more formally and in the end of the introduction.

Spell out  BT#9 (if an extended name exists)

Considering the sentence “Target specificity of this lead compound….” indicate if there is lead “Pb” in this compound, since in the figure with the molecular structure of BT#9 there is no Pb.

In the text “Results” each time a difference is mentioned as significant, “(p<0.05)” should be indicated.

The authors demonstrate BT#9-induced augmented mitochondrial ROS which leads to cell necrosis; it could also be interesting to discuss another ensuing aspect: how tissue necrosis leads to inflammation with subsequent oxidative stress. It beneficial (or not) significance in therapeutic terms should be considered.

Author Response

We very appreciate your review and recommendations to improve out manuscript. Please find our responses in the attachment.  We have incorporated the changes in the revised manuscript.

Thank you.

Reviewer 2 Report

Authors have evaluated an approach for targeted inhibition of mitochondria-associated granulocyte-macrophage colony-stimulating factor signaling (Magmas) in prostate cancer. Authors show promising insights into the inhibition of magmas using synthesized BT#9 molecule, although further in vivo studies needed.

The overall manuscript is well organized, and experimental designs were appropriate to study the efficacy of inhibitor molecule supported by functional assays.

Minor comments

Please include the description of samples/data sets with accession numbers used from oncomine/cBioPortal databases as supplementary data for the reproducibility of analyses performed.

Resolutions for Figs. 5 B & D and Figs. 6 B, D & F needs to be improved, and figures need to be aligned.

All the raw data used for statistical analyses with P-values should be made available as supplementary tables.

Authors may include a discussion on observations related to magmas through their cancer dependency maps, and any potential side effects.

Author Response

We very much appreciated your review and the recommendations to improve our manuscript.   Please see the attachment and we have incorporated the changes with in the manuscript and supplementary file.

Thank you.
